# Advanced Paternal Age and Sperm Proteome Dynamics: A Possible Explanation for Age-Associated Male Fertility Decline

**DOI:** 10.3390/cells14110813

**Published:** 2025-05-30

**Authors:** Joana Santiago, Joana V. Silva, Manuel A. S. Santos, Margarida Fardilha

**Affiliations:** 1Department of Medical Sciences, Institute of Biomedicine, University of Aveiro, 3810-193 Aveiro, Portugal; joanavieirasilva@ua.pt (J.V.S.); mfardilha@ua.pt (M.F.); 2Multidisciplinary Institute of Ageing, MIA-Portugal, University of Coimbra, 3000-370 Coimbra, Portugal; mansilvasantos@uc.pt

**Keywords:** advanced paternal age, male infertility, protein, phosphorylation, spermatozoa

## Abstract

Male fertility is strongly influenced by environmental exposures, lifestyle, and advancing age. While advanced paternal age (APA) has been linked with a progressive decline in male fertility, poor reproductive outcomes, and decreased offspring health, the molecular mechanisms underlying these alterations remain unclear. In this work, we investigated the impact of men’s age on human sperm protein expression and phosphorylation to identify molecular alterations possibly responsible for the age-associated decline in male fertility. Semen samples from volunteers attending fertility consultations at the Hospital of Aveiro were collected, analyzed according to WHO’s guidelines, and processed by the density gradient technique. The proteome and phosphoproteome of 19 normozoospermic human sperm samples divided into four age groups were evaluated by mass spectrometry: ≤30 years old; 31–35 years old; 36–40 years old; and >40 years old. Proteomic analysis revealed 46 differentially expressed proteins (DEPs) between groups, some of them associated with infertility-related phenotypes. Gene ontology (GO) analysis, performed using the DAVID database, revealed that DEPs in older men were enriched in pathways related to stress response, metabolism, and embryo implantation. Additionally, 94 differentially phosphorylated sites corresponding to 76 differentially expressed phosphorylated proteins between the groups were identified, related to key reproductive processes such as sperm motility, spermatogenesis, and sperm binding to zona pellucida, and involved in metabolic and stress response pathways, like HSF1 activation. The set of proteins and phosphorylated residues altered in the sperm fraction usually used in assisted reproductive technology (ART) highlights the need to consider the age of the male partner during fertility assessment and treatment planning. These markers can also be used to explain cases of idiopathic infertility, failure in ART, or repeated abortion associated with APA, overcoming the subjectivity of the conventional semen analysis.

## 1. Introduction

Infertility is a pressing issue in modern societies, especially in industrialized regions where there has been a significant decrease in childbirths over the decades [1,2]. This problem affects around 15% of couples of reproductive age, leading to emotional distress for the couple and presenting broader societal implications [1]. A significant contributing factor is the growing trend to delay parenthood until later stages of life for both partners, with the average maternal age at first child now exceeding 30 years in most countries, including Portugal [3]. While personal choices and socioeconomic factors often contribute to this trend, lifestyle-related issues such as obesity, sedentary lifestyle, and unhealthy habits among reproductive-aged individuals further perpetuate the problem.

Although women experience a pronounced decline in fertility with age [4,5], in men, a gradual decline in reproductive capacity rather than a complete cessation of fertility typically occurs [6]. Despite idiopathic infertility remaining the most common type of male infertility, advanced age has been hypothesized as an important cause of sperm quality decline [2]. Most studies investigating the age-associated male reproductive decline indicate that age negatively affects basic semen parameters, although an absent effect was also reported [7,8]. Moreover, a concerning link between advanced paternal age (APA) and negative reproductive outcomes has emerged, such as lower pregnancy rates, increased risk of pregnancy loss, and a higher incidence of developmental, morphological, and neurological disorders in newborns [9,10,11]. These conditions, ranging from achondroplasia, schizophrenia, and autism to autosomal dominant disorders, aneuploidies, and cancers, raise significant concerns about not only how the natural aging process affects male fertility but also what the true associated risks and consequences are for offspring. These concerns are amplified by the widespread use of assisted reproductive techniques (ART), which usually devalues the age of the male partner. Despite the definition of APA remaining a topic of debate, it typically refers to men who are older than 40 years at the time of conception [11].

The molecular mechanisms underlying age-associated male fertility decline are still not fully understood. Aging has been associated with molecular changes in sperm, including DNA damage and chromosomal abnormalities [12], oxidative stress [13], mitochondrial dysfunction [14], and epigenetic changes [11]. Epigenetic modifications, including DNA methylation, histone modifications, and small non-coding RNAs, have been highlighted as key modulators of fertility, embryo development, and offspring’s health [11]. In addition to these, proteins also seem to play crucial roles in the development of the preimplantation embryo [15], besides their already recognized involvement in sperm motility, capacitation, acrosome reaction, and post-fertilization events [16]. How protein expression in sperm changes with aging is still poorly explored. To date, only two studies have reported alterations in protein content in the sperm of aged men using mass spectrometry [17,18]. Globally, they showed that alterations in sperm proteome were associated with the decline of semen quality in aged males and may serve as biomarkers for predicting semen quality. Furthermore, considering that transcription and translation are residual biological processes in sperm, sperm function is mainly regulated by post-translational modifications (PTMs), including protein phosphorylation [19,20]. In the past years, studies investigating the alteration of sperm phosphoproteome associated with capacitation [21,22] and sperm motility [23,24] emerged. However, to the best of our knowledge, no information concerning the change in sperm phosphoproteome with age is available.

Considering the little known regarding the molecular changes in sperm caused by the aging process and their possible implications for male fertility decline, in this study, we aimed to fill this gap by investigating the impact of men’s age on human spermatozoa proteins and phosphoproteins using proteomics. The recognition of altered protein expression and phosphorylation in men with advanced age allowed the identification of signaling pathways that may underlie age-related reproductive decline in men, even when conventional semen parameters remain within normal limits.

## 2. Materials and Methods

### 2.1. Sample Collection

This study was approved by the Ethics and Internal Review Board of the Hospital Infante D. Pedro E.P.E. (Aveiro, Portugal) (Process number: 36/AO; Approved on 14 April 2015) and by the Ethics Committee of Centro Hospitalar Vila Nova de Gaia/Espinho, E.P.E. (Vila Nova de Gaia, Portugal) (Process number: 12/2019-3; Approved on 18 July 2019) and was conducted following the ethical standards of the Declaration of Helsinki. All donors signed an informed consent allowing the use of samples for scientific purposes. Exclusion criteria included any known conditions affecting male reproductive system and/or male fertility (varicocele, cryptorchidism, orchitis, epididymitis, endocrine hypogonadism, obstruction of the vas deferens), as well as ongoing medication, smoking, and heavy alcohol consumption. Ejaculated semen samples were obtained from donors recruited at the Hospital Infante D. Pedro E.P.E. (Aveiro, Portugal) between January 2019 and December 2020 by masturbation into a sterile container after 2 to 7 days of sexual abstinence. Basic semen analyses were conducted following the 5th edition of the World Health Organization’s guidelines [25].

### 2.2. Density Gradient Centrifugation

To rule out the possibility of contamination by somatic cells and debris and to enrich the sample in highly motile sperm, a density gradient sperm selection was performed. This type of semen treatment is routinely performed in ART procedures to isolate the best sperm and increase the chances of fertilization. Sperm cells were washed after semen liquefactionby the density gradient method using SupraSperm^®^ (Origio, Ballerup, Denmark) according to the manufacturer’s instructions. Briefly, 90% and 45% gradients were prepared using SupraSperm^®^ 100 and Sperm Preparation Medium (Origio, Denmark) and pre-equilibrated in a CO_2_ environment at 37 °C; 1 mL of each gradient was used per each 2 mL of the liquefied semen sample. The gradient was centrifuged at 300× *g* for 20 min at room temperature (RT), and the pellet was washed twice with Sperm Preparation Medium (Origio, Denmark) at 300× *g* for 10 min, and the motility and concentration of spermatozoa in the washed sample were determined. Optical phase contrast microscopic examination was used to confirm the absence of somatic cells. The viable fractions of spermatozoa were cryopreserved using CryoSperm™ (Origio, Denmark) according to the manufacturer’s instructions and stored at −80 °C until used for subsequent experiments.

### 2.3. Protein Extraction and Trypsin Treatment

Nineteen randomly selected normozoospermic sperm samples were divided into four groups according to men’s age: (G1) ≤30 years; (G2) 31–35 years; (G3) 36–40 years; and (G4) >40 years; and prepared for liquid chromatography-tandem mass spectrometry (LC-MS/MS) analysis (Appendix A). Cell pellets were homogenized in 200 µL lysis buffer containing 25 mM Hepes pH 7.5, 25 mM NaCl, 0.4% NP-40, 2 mM DTT, protease inhibitor cocktail tablet, 0.4% Amphipol A8-35 (Anatrace, cat. No. A835). Samples were vortexed vigorously, incubated at 4 °C for 30 min, and 3 cycles of freezer–thawing were performed using liquid nitrogen. Then, the samples were diluted 4 times with 25 mM Hepes pH 7.5 and 25 mM NaCl and sonicated with 3 pulses of 15 s at an amplitude of 20% using a 3 mm probe, with incubation on ice for 30 s between pulses. Debris was pelleted by centrifugation at 20,000× *g* for 20 min at 4 °C, the protein concentration of the supernatant was measured by bicinchoninic acid (BCA) assay (Thermo Scientific, Waltham, MA, USA), and from each sample, 100 µg protein was isolated to continue the protocol. Proteins were reduced by the addition of 15 mM dithiothreitol and incubation for 30 min at 55 °C and then alkylated by the addition of 30 mM iodoacetamide and incubation for 15 min at RT in the dark. The samples were acidified with 10% formic acid (FA) to pH 3 and centrifuged for 10 min at 15,000× *g* and RT. The resulting protein-containing pellets were dissolved in 50 µL 50 mM triethylammonium bicarbonate (TEAB), and proteins were digested overnight with 1 µg trypsin (1:100, *w*:*w*, Promega) at 37 °C. Samples were acidified with trifluoroacetic acid (TFA) to a f.c. of 1% (pH < 3), incubated for 15 min on ice, and centrifuged for 10 min at 20,000× *g* and RT to pellet the precipitated Amphipol A8-35. For shotgun analysis, 5 µg clean peptides were isolated for each sample and dried completely by vacuum centrifugation. The remainder was used for phosphopeptide analysis.

### 2.4. Phosphopeptides’ Enrichment

Phosphopeptides were enriched with MagReSyn^®^ Ti-IMAC beads according to the manufacturer’s instructions with minor modifications. Briefly, 20 µL MagReSyn^®^ Ti-IMAC beads per sample were washed twice with 70% EtOH, once with 1% NH_4_OH, and three times with a mixture of water/ACN/TFA (14:80:6, *v*/*v*/*v*). Next, the digested sample was incubated with the washed beads for 30 min at RT. The beads were washed once with a mixture of water/ACN/TFA (14:80:6, *v*/*v*/*v*) and three times with a mixture of water/ACN/TFA (19:80:1, *v*/*v*/*v*). Phosphopeptides were eluted from the beads by adding three times 80 µL 1% NH_4_OH. An amount of 60 µL 10% FA was added to the combined eluate, and the samples were dried completely by vacuum centrifugation. QCloud was used to control instrument longitudinal performance during the project [26,27].

### 2.5. LC-MS/MS Analysis

Each sample was solubilized in 20 µL loading solvent A (0.1% TFA in water:ACN (98:2, *v*:*v*)) moments before analysis. For the shotgun analysis, 2 µg of the sample measured on Dropsense16 (Unchained Labs, Pleasanton, CA, USA) was injected for LC-MS/MS analysis on an Ultimate 3000 RSLCnano system in-line connected to an Orbitrap Fusion Lumos mass spectrometer (Thermo Fisher Scientific, Waltham, MA, USA). For the PTM analysis, 15 µL of the sample was injected for LC-MS/MS analysis on an Ultimate 3000 RSLCnano system in-line connected to an Orbitrap Fusion Lumos mass spectrometer (Thermo Fisher Scientific, Waltham, MA, USA). Trapping was performed at 10 μL/min for 4 min in loading solvent A on a 20 mm trapping column (made in-house, 100 μm internal diameter (I.D.), 5 μm beads, C18 Reprosil-HD, Dr. Maisch, Ammerbuch, Germany). The peptides were separated on a 50 cm µPAC™ column with C18-endcapped functionality (prototype, Thermo Scientific). It was kept at a constant temperature of 50 °C. Peptides were eluted by a linear gradient reaching 22.5% MS solvent B (0.1% FA in water/acetonitrile (2:8, *v*/*v*)) after 109 min, 30.5% MS solvent B at 135 min, 55% MS solvent B at 153 min, 70% MS solvent B at 155 min, followed by a 5 min wash at 70% MS solvent B and re-equilibration with MS solvent A (0.1% FA in water). The flow rate was set to 250 nL/min. The mass spectrometer was operated in data-dependent mode. Full-scan MS spectra (300–1500 m/z) were acquired at a resolution of 120,000 in the Orbitrap analyzer after accumulation to a target AGC value of 200,000 with a maximum injection time of 50 ms. Precursor ions were filtered for charge states (2–7 required), dynamic exclusion (60 s; +/− 10 ppm window), and intensity (minimal intensity of 5 × 10^4^). The precursor ions were selected in the quadrupole with an isolation window of 1.2 Da and accumulated to an AGC target of 1.2 × 10^4^ or a maximum injection time of 40 ms for the shotgun analysis and 100 ms for the PTM analysis and activated using CID fragmentation (34% NCE). The fragments were analyzed in the Ion Trap Analyzer at a turbo scan rate.

### 2.6. LC-LC/MS Data Analysis

Mass spectrometry data analysis was performed in MaxQuant (version 2.0.1.0) with mainly default search settings, such as a false discovery rate (FDR) set at 1% on peptide-to-spectrum matches (PSM) at, peptide and protein levels. Spectra were searched against the human reference proteome (version of 2021_01, UP000005640). The mass tolerance for precursor and fragment ions was set to 4.5 and 20 ppm, respectively, during the main search. Enzyme specificity was set as C-terminal to arginine (Arg/R) and lysine (Lys/K), also allowing cleavage at proline (Pro/P) bonds with a maximum of two missed cleavages. Variable modifications were set to oxidation of methionine (Met/M) residues, acetylation of protein N termini and phosphorylation of serine (Ser/S), threonine (Thr/T), and tyrosine (Tyr/Y). Fixed modification was set to carbamidomethylation on the cysteine (Cys/C) residues. A matching time window of 0.7 min and an alignment time window of 20 min was used to reach the matching between runs. Only proteins with at least one unique or razor peptide were retained. Proteins were quantified by the MaxLFQ algorithm integrated into the MaxQuant software v2.6.8.0. A minimum ratio counts of two unique or razor peptides was required for quantification. For the shotgun analysis, a total of 396,209 PSMs were performed, leading to 17,734 identified unique peptides, which correspond to 2330 identified protein groups (Appendix A). Further data analysis of the shotgun results was performed with an in-house R script, using the protein groups output table from MaxQuant. Reverse database hits were removed, LFQ intensities were log2 transformed, and replicate samples were grouped. Proteins with less than three valid values in at least one group were removed, and missing values were imputed from a normal distribution centered around the detection limit (package DEP) [28], leading to a list of 1214 quantified proteins in the experiment, used for further data analysis (Appendix A). To compare protein abundance between pairs of sample groups, statistical testing for differences between two group means was performed using the package limma [29] (Appendix A). For the PTM analysis, a total of 9502 phosphopeptides were identified, corresponding to 1566 phosphorylation sites (Appendix A). These 1566 phosphosites were derived from 1146 different phosphopeptides. Further data analysis of the PTM results was performed with an in-house R script using the phosphor (STY) sites output table from MaxQuant. Intensity values were log2 transformed, and replicate samples were grouped, leading to a list of 300 phosphopeptides, corresponding to 298 unique phosphosites on 197 proteins (Appendix A). To compare phosphopeptide abundance between pairs of sample groups, statistical testing for differences between two group means was performed using the package limma [29] (Appendix A). Statistical significance for differential regulation was set to a *p*-value of <0.05 and fold change of >4.5- or <0.58-fold (|log2FC| = 1.5).

### 2.7. Bioinformatic Analysis

The UniProt database was used to retrieve involvement in diseases and gene ontology (GO) information of the differentially expressed proteins (DEPs) and phosphorylated proteins (DEPPs) identified (data were downloaded on 9 March 2023) (Appendix A). To gain molecular insight into the DEPs and DEPPs identified, a GO enrichment analysis was performed for biological processes, molecular functions, and cellular components using DAVID Bioinformatics Resources (v22q4) [30,31] (downloaded on 24 May 2023). The REACTOME database was used to identify signaling pathways associated with the DEPs and DEPPs identified in each group based on available literature (accessed on 24 May 2023). Only terms with a *p*-value < 0.05 were retrieved. The database Disease (accessed on 9 March 2023) was explored to identify DEPs previously associated with male infertility phenotypes. The DEPs and DEPPs identified were searched in the PubMed database (Bethesda, Rockville, MD, USA) to investigate their role in mammalian spermatozoa and fertilization (search performed until June 2023). Human protein–protein interaction (PPI) data was retrieved from the HIPPIE and STRING databases (downloaded on 10 October 2023). The HIPPIE database is regularly updated by incorporating interaction data from major expert-curated experimental PPI databases (such as Bell09, BioGRID, HPRD, IntAct, and MINT). The STRING database integrates all known and predicted associations between proteins, including both physical interactions and functional associations. Network analyses were performed using Cytoscape (version 3.9.1; Bethesda, MD, USA). The PhosphoSitePlus^®^ website was searched to identify known and novel phosphosites identified by this analysis (accessed on 30 May 2023). PhosphoSitePlus^®^ is an online database that compiles comprehensive information and tools for the study of PTMs, including phosphorylation, based on mass spectrometry experiments [32].

### 2.8. Protein Extraction and Slot Blot

For total protein extraction, 5 µL of Sodium Dodecyl Sulfate 1% (SDS 1%) (per million spermatozoa was added to the sperm pellet and incubated for 5 min with agitation. Lysates were centrifuged for 15 min at 4 °C at 16,000× *g*, and the supernatants (soluble fraction) were collected and stored at −30 °C. The Thermo Scientific™ Pierce™ BCA Protein Assay Kit (Fisher Scientific, Loures, Portugal) was used to determine the total protein concentration of the samples according to the manufacturer’s instructions. Samples were diluted to 0.05 μg/μL and blotted under vacuum into a nitrocellulose membrane, 0.45 μm pore size (GE Healthcare, Chicago, IL, USA), inside the slot blot device (BioRad Portugal, Sintra, Portugal). The membranes were blocked using 5% (*w*/*v*) nonfat dry milk in tris-buffered saline (TBS) with 0.1% Tween™ 20 (TBS-T) for 1 h at RT. Incubation with the primary antibodies occurred for 1 h at RT using the following dilutions: mouse anti-LYZL1 antibody (1:450, B01P, Abnova, Taipé, Taiwan), mouse anti-LAMP1 (1:1000, #15665, Cell Signaling, Danvers, MA, USA), mouse anti-HSF1 (1:500, sc-177757, Santa Cruz Biotechnology, Dallas, TX, USA), mouse anti-HSP90 (1:2000, 13171-1-AP, Proteintech, Manchester, UK), mouse anti-HSP27 (1:1000, sc-13132, Santa Cruz Biotechnology, Dallas, TX, USA), mouse anti-p-HSP27 (Ser82) (1:1000, sc-166693, Santa Cruz Biotechnology, Dallas, TX, USA). After being washed in TBS-T, membranes were incubated with IRDye^®^ 800CW goat anti-mouse secondary antibody (1:10,000) for 1 h at RT. Membranes were scanned using the Odyssey Infrared Imaging System (LI-COR^®^ Biosciences, Lincoln, NE, USA). Results were normalized to total protein levels determined by Ponceau staining, which was performed before antibody probing.

### 2.9. Statistical Analyses

Descriptive statistics of all data were calculated using RStudio Version 1.2.5033. A Kruskal–Wallis test was used to detect differences between age groups. For the determination of correlations between age and the levels of LAMP1, LYZL1, HSF1, HSP90, HSP27, and p-HSP17, and the Spearman’s correlation test was applied after the evaluation of normality using the Shapiro–Wilk test. The significance level was set at 0.05. All analyses were conducted using RStudio Version 1.2.5033.

## 3. Results

### 3.1. Sperm from Men of Advanced Age Present Alterations in the Proteome

#### 3.1.1. Identification of Differentially Expressed Proteins

To identify differences in protein expression and phosphorylation in the sperm of men with APA, the sperm fraction enriched in highly motile and viable sperm of 19 normozoospermic semen samples was analyzed using LC-MS/MS (Appendix A). No significant differences were observed regarding seminal parameters between groups (Table 1).

Among the 2330 protein groups identified (Appendix A), 1214 were reliably quantified, meaning protein groups with at least three valid LFQ intensity values in one of the experimental conditions and used for differential expression analyses (listed in Appendix A). The proteomic analysis revealed 46 DEPs between the groups (Table 2 and Figure 1A), including 10, 12, and 1 DEPs between G1 (<30 years old) and G2 (31–35 years old), G3 (36–40 years old), and G4 (>40 years old), respectively. Sixteen DEPs were identified between G2 and G3, and 13 between G2 and G4. Finally, 11 DEPs were found between G3 and G4. In total, 25 proteins were differentially expressed in G4 compared with the other age groups, comprising 13 upregulated proteins and 12 downregulated.

#### 3.1.2. Functional Analysis of Differentially Expressed Proteins Between Groups

All DEPs were previously identified in the human sperm proteome [33] except TP53I11, MT-CO3, and DEFA3. The proteins PLCZ1, TEX101, PRSS37, LRGUK, CFAP54, IZUMO3, TMCO5A, and IQCF5 were associated with male infertility according to the DISEASE database. TP53I11 and EDDM3B were also associated with Sertoli cell-only syndrome and TEX101, EDDM3B, and PLCZ1 with varicocele. Additionally, EDDM3B and LYZL1 were linked to spermatogenic failure, EDDM3B and CCDC90B with azoospermia, and EDDM3B with oligoasthenoteratozoospermia.

GO analysis revealed the significant biological processes associated with DEPs across all age groups (*p*-value < 0.05) (Figure 1B and Appendix A), which included negative regulation of phosphatase activity (GO:0032515) in G1 and G2, cellular respiration (GO:0045333) and mitochondrial electron transport, cytochrome c to oxygen (GO:0006123) in G2, G3, and G4 (Figure 1B and Appendix A). Response to unfolded protein (GO:0006986), positive regulation of the apoptotic process (GO:0043065), embryo implantation (GO:0007566), and response to heat (GO:0009408) are biological processes significantly affected in the sperm of men with APA (G4) (Figure 1B and Appendix A). Concerning the molecular mechanisms, cytochrome-c oxidase activity (GO: 0004129) was the most significantly enriched in G2, G3, and G4, and unfolded protein binding (GO:0051082) and ATPase activator activity (GO:0001671) in G3 and G4 (Figure 1B and Appendix A). The most enriched pathways across all groups were cellular response to stress and cellular response to stimuli (Figure 1B and Appendix A). Other significant pathways included respiratory electron transport, metabolism, and response to chemical stress in G2, G3, and G4 (Figure 1B and Appendix A).

### 3.2. Advanced Paternal Age Is Associated with Alterations in the Sperm Phosphoproteome

#### 3.2.1. Identification and Quantification of Phosphopeptides in Human Sperm Processed by Density Gradient Centrifugation

The phosphoproteomic analysis identified 9502 phosphopeptides, corresponding to 1566 phosphorylation sites derived from 1146 different phosphorylated proteins (Appendix A). Among them, 300 phosphopeptides were reliably quantified, meaning phosphopeptides with at least three valid intensity values in one of the experimental conditions, which correspond to 298 unique phosphosites on 197 proteins (Appendix A). To characterize the distribution of phosphosites, the number of modification sites per phosphorylated protein was evaluated. The results showed that most proteins contained one (73%), two (17%), or three (6%) phosphosites (Figure 2A). AKAP3 and AKAP4 exhibited the highest number of phosphorylation sites, with 10 and 20 phosphorylation sites, respectively. Most phosphorylation events occurred on Ser residues (89%), followed by Thr (8%) and Tyr (3%) (Figure 2B). The PhosphoSitePlus^®^ website was searched to identify known and novel phosphosites identified by this analysis. The results revealed 145 novel phosphosites (49%) mapped to 100 different proteins (Appendix A).

#### 3.2.2. Differences in Phosphorylated Proteins Between Age Groups

A total of 94 differentially phosphorylated sites corresponding to 76 differentially expressed phosphorylated proteins (DEPPs) between the groups were identified (*p*-value < 0.05 and |log2FC| = 1.5) (Table 3). The number of shared differentially expressed phosphosites (Figure 2C) and phosphorylated proteins (Figure 2D) among the groups were investigated, revealing that the phosphorylation levels of AKAP4 (S245 and S447), C7orf61 (S34), KNG1 (S332), and RPL13 (S106) were altered in G1 compared with all the older age groups, while the phosphorylation of ADAM21 at S670 and PGK2 at S174 was decreased in G4 compared with all the younger age groups. Also, seven phosphorylated proteins were deregulated in G1 compared with G3 and G4 (AKAP4, DNAH8, HSP90AB1, ODF2, PDHA1, MTCH2, and TRIML1) and six between G2 and the older groups (FAM209B, ADGRF1, ATP8B3, TSNAXIP1, PSMA6, and SPATA6) were found. The phosphorylation levels of RPL13 (S106), PGK2 (S174), and TRIML1 (S67) were the most significantly upregulated (FDR < 0.1 and |log2FC| = 1.5) in G1 compared with G4 (Figure 3A). Additionally, G2 presented two phosphosites (S175 and S2 at PSMA6) significantly upregulated (FDR < 0.1 and |log2FC| = 1.5) compared with G3 (Figure 3B) and 1 (S2 at PSMA6) compared with G4 (Figure 3C).

#### 3.2.3. Functional Analysis of Differentially Expressed Phosphorylated Proteins Between Groups

GO enrichment analysis was performed to identify biological processes, molecular functions, cellular components, and pathways significantly associated with the altered phosphorylated proteins (Appendix A). The most significantly enriched biological processes in all groups were flagellated sperm motility (GO:0030317) and spermatogenesis (GO:0007283) (Figure 3D). The DEPPs in G4 were also significantly involved in processes such as chaperone-mediated protein complex assembly (GO:0051131), cell differentiation (GO:0030154), positive regulation of telomerase activity (GO:0051973), and binding of sperm to zona pellucida (GO:0007339). Across all groups, DEPPs were significantly associated with the sperm connecting piece, sperm midpiece, nucleus, fibrous sheath, and extracellular exosome (Figure 3D). The most common molecular functions were protein binding and unfolded protein binding, ATPase activity, and pyruvate dehydrogenase activity (Figure 3D). REACTOME pathway enrichment of the DEPPs showed altered phosphorylation levels in proteins enriched in HSF1 activation, G2/M transition, mitotic G2-G2/M phases, RHOBTB GTPase cycle, signaling by Hedgehog, among many others (Figure 3D).

### 3.3. Integrative Network of Altered Proteins and Phosphorylated Proteins in Sperm of Men Older than 40 Years

To better understand the possible interaction between the altered proteins and the changes in protein phosphorylation, a network was constructed with the DEPs and DEPPs in the G4 compared with the other three groups (Figure 4). The total number of proteins (square nodes) in the PPI network was 25, with 13 proteins upregulated in G4 (green outline in Figure 4) and 12 downregulated (red outline in Figure 4), compared withG1 (yellow nodes), G2 (blue nodes), and/or G3 (pink nodes). The network also included the 60 DEPPs in the G4 compared with the other three groups (round nodes in Figure 4) and the known kinases that regulate their phosphorylation (diamond nodes). The proteins AKAP3 and AKAP4, PSMA6, and VCP presented phosphorylated residues both up- and downregulated (dark blue outlines in Figure 4). According to the PhosphoSitePlus^®^ database, casein kinase 2 alpha 1 (CK2A1) is responsible for the phosphorylation of HSP90BA1 at S226, AKT serine-threonine kinase 1 (AKT1) for RPL13 phosphorylation at S106, and serine/threonine-protein kinase PAK1 for ARL4A phosphorylation at S143 (Figure 4).

The total number of DEPs and DEPPs in the PPI network is N = 88, and the total number of interactions between them was L = 92. Eight DEPs and 26 DEPPs from the dataset have no PPI data available. The average connectivity, defined as the average number of neighbors per node in the network, was 3.955, and the mean clustering coefficient, which indicates how the nearest neighboring nodes of a node are connected to each other, was C = 0.265. The more the nodes are connected to each other, the closer the clustering coefficient will be to 1. On the opposite, a sparse random uncorrelated network presents a coefficient close to 0. Therefore, considering the relatively high clustering coefficient of this network, with a structure with highly connected proteins, we can hypothesize that the DEPs and DEPPs are related and, collectively, may play a common functional role in situations of advanced age in sperm.

#### Validation of Identified Proteins by Slot Blot

Two lysosomal proteins were selected for validation based on their expression pattern and their potential relevance to male fertility and fertilization: (i) LAMP1, downregulated in the sperm of men older than 35 years old (G3 and G4) compared with men younger than 30 years old (G1); and (ii) LYZL1, downregulated in men aged between 36 and 40 years old (G3) compared with men younger than 35 years old (G1 and G2). The levels of these proteins were evaluated by slot blot in a new cohort of 48 sperm samples with distinct semen parameters, processed by density gradient centrifugation (Appendix A). A weak correlation between the levels of LYZL1 and men’s age was observed (r = −0.294, *p*-value = 0.042) (Figure 5A). No correlation was found between the levels of LAMP1 and men’s age (r = −0.138, *p*-value = 0.3482) (Figure 5B).

Considering that GO enrichment analysis revealed the involvement of DEPs and DEPPs in the cellular response to stress and stimuli, including the HSF1 activation pathway, we evaluated the levels of some key proteins involved in this pathway—HSF1 (Figure 5C), HSP90 (Figure 5D), HSP27 (Figure 5E), and phosphorylated HSP27 (Figure 5F). This analysis revealed a weak negative correlation between the levels of HSP27 and age (r = −0.299, *p*-value = 0.039) (Figure 5E) but a weak positive correlation between its phosphorylation levels and age (r = 0.257, *p*-value = 0.078) (Figure 5F). Furthermore, no significant correlation was observed between age and the levels of HSF1 (r = −0.141, *p*-value = 0.338) (Figure 5C) or HSP90 (r = −0.122, *p*-value = 0.408) (Figure 5D). No significant differences in the levels of these proteins were observed across the groups, except for phosphorylated HSP27 levels, which were significantly increased in G3 (mean 22.77 ± 9.40) compared with G1 (mean 6.06 ± 1.38) (Appendix A).

## 4. Discussion

In the past decades, increased attention has been directed toward the effect of APA on male reproductive capacity, reproductive outcomes, and offspring health. Indeed, the widespread use of ART, without clear limitations concerning men’s age, has raised many concerns about the safety of using sperm from older men and the possible consequences for future generations [34]. Despite growing interest, the mechanisms by which APA exerts adverse effects on semen quality, ART outcomes, and offspring health remain largely unknown. Thus, elucidating the molecular characteristics of aging sperm is essential.

In this study, we conducted a comprehensive quantitative proteomic and phosphoproteomic analysis of highly motile and viable sperm from normozoospermic men stratified by age. The use of density gradient centrifugation ensured the isolation of sperm typically associated with better pregnancy rates after ART treatment [35,36]. Forty-six DEPs between the groups were identified (Table 2), 25 of them with altered expression in men with APA (G4) compared with the other groups (Figure 4). To date, only two studies have reported alterations in sperm protein composition with age using a similar approach [17,18]. Liu et al. compared the proteomic profile of young adults (28–32 years old) and aged men (68–72 years old) and found thirteen downregulated and nine upregulated proteins in the older group, the majority related to structural, transport, and chaperone functions; enzymatic activity; signal transduction; and reproduction [17]. None of the identified proteins also appeared in our study, which may be explained by the different age groups used, especially since we did not include such old men. Recently, Guo et al. identified 80 DEPs between young men (26–36 years old) and older men (46–54 years old), which were significantly involved in protein digestion and absorption, ubiquitin-mediated proteolysis, and energy-related pathways such as oxidative phosphorylation [18]. Notably, two common DEPs were identified, PLCZ1 and TFAM, previously found overexpressed in spermatozoa of men aged between 46 and 54 years old [18]. However, despite we also found higher levels of TFAM in G4 (>40 years old) compared with G2 (31–35 years old), PLCZ1 was downregulated in G3 (36–40 years old) compared with G2 (31–35 years old). PLCZ1 was described as an oocyte-activating factor in mice by triggering Ca^2+^ oscillations like those occurring following sperm penetration [37,38]. Its downregulation may indicate poor fertilization success, similar to what occurs in situations of *PLCZ1* mutations [38,39,40].

We also found age-related decline in LAMP1, LRGUK, and LYZL1, proteins involved in acrosome function and sperm maturation. LAMP1 is a glycoprotein that plays an important role in lysosome biogenesis and autophagy, being present in the lysosomal membrane of the acrosome in mammalian sperm. LAMP1 appears late in spermatogenesis during the acrosome phase, apparently associated with cytoplasmic vesicles in the residual body [41]. LYZL1 is located in the sperm head and also in the acrosome matrix in rodents [42,43] and was previously linked to spermatogenic and fertilization failure since this protein is required for proper acrosome reaction in rats [44]. It also presents muramidase, isopeptidase, and antibacterial activities, supporting a role in male reproductive tract immunity [43]. The downregulation of LAMP1 and LYZL1 in older patients was further confirmed in 48 sperm samples with distinct semen parameters, supporting a progressive decline in acrosomal protein expression with age (Figure 5). However, the role of LAMP1 and LYZL1 in fertilization, in particular its involvement in capacitation, acrosome reaction, and digestion of the zona pellucida, remains unclear, and future studies should focus on this hypothesis.

Proteins involved in metabolism and mitochondrial function were also differentially expressed in men older than 35 years compared with the sperm of younger men—COX4I1, AK1, BSG, TP53I11, LPP, and IQCF5. The three downregulated proteins COX4I1, AK1, and BSG were associated with metabolic pathways, suggesting impaired mitochondrial respiration and ATP synthesis in older sperm. Indeed, COX4I1 is involved in cellular respiration and mitochondrial electron transport (cytochrome c to oxygen), two biological processes significantly linked to the DEPs identified (Figure 1). The main energy source driving sperm motility is ATP, mainly derived from glycolysis and oxidative phosphorylation [45,46]. However, the metabolic demands of the sperm may require the use of power stored in high-energy phosphate bonds [45]. AK1 is a phosphotransferase associated with the outer microtubular doublets and ODFs of the midpiece and principal piece of the sperm flagellum, which in situations of high energy consumption catalyzes the reaction 2ADP ↔ ATP + AMP, generating the energy required for sperm function [45,47]. Despite *Ak1* knockout mice not showing any defects in testis development, spermatogenesis, sperm morphology, and motility under physiological conditions, they presented compromised sperm motility (smaller beating amplitude and higher beating frequency) that resulted in less effective forward swimming when only ADP was available [48]. This supports the hypothesis that aging sperm may activate energy pathways, such as the AK1-mediated phosphotransferase system, to compensate for deficits in oxidative phosphorylation and glycolysis.

The identification of altered phosphorylation patterns in specific conditions (e.g., pathology, exposure to chemicals, risk factors) is important to unravel their influence on protein kinases and phosphatase activity and identify the signaling pathways affected. In sperm, this is particularly important since this highly specialized type of cell is almost absent of transcription and translation, relying on PTMs and PPIs for regulating its function [19,49]. In this study, we used phosphoproteomics to investigate for the first time the alterations in the human sperm phosphoproteome with age. We identified 1566 phosphorylation sites on 1146 proteins. A previous study on human sperm reported a total of 3527 phosphorylation sites on 1322 proteins, of which the phosphorylated proteins with the most modification sites were AKAP4 and AKAP3, similar to our findings [20]. Few Tyr phosphorylation sites (3%) were identified compared with phosphorylation in Ser (89%) and Thr (8%), similar to what was found in bovine sperm and closer to what usually occurs in somatic cells, in which it is estimated that the proportion of the occurrence of Ser, Thr, and Tyr phosphorylation is 1000:100:1 [49] (Figure 2). Interestingly, a large number of novel phosphorylated sites in 100 proteins were identified in this study (49%), questioning their specificity and relevance for sperm function.

The phosphoproteomic analysis revealed 94 altered phosphorylation sites mapped to 76 DEPPs between the age groups (Table 3). The association of these DEPPs with specific sperm structures, including the connecting piece, midpiece, and fibrous sheath (Figure 3), suggests their involvement in sperm-specific functions such as sperm motility and fertilization. This was further supported by GO analysis, which revealed that spermatogenesis, flagellated sperm motility and binding of sperm to zona pellucida were among the most significantly enriched biological processes (Figure 3D). Indeed, previous studies showed that spermatogenesis is affected by age, supported by the loss of germ cells observed in older men and, thus, lower daily sperm production [6]. Additionally, the phosphorylation at S174 of the testis-specific isozyme PGK2, essential for sperm motility and male fertility, was also decreased in the sperm of men with APA. The ablation of this enzyme, which catalyzes the first step of ATP generation in the glycolytic pathway, resulted in a severe reduction in sperm motility and ATP levels in mice, while testis histology, sperm counts, and morphology remained unchanged [50]. Previous reports have shown that PGK2 expression not only significantly decreases in sperm of idiopathic asthenozoospermic patients [46,51], normozoospermic men consulting for infertility and normozoospermic men with IVF failure [52], but also declines in the tests and spermatozoa of elderly men [46,53]. However, how the decrease in phosphorylation levels impacts PGK2 activity and affects sperm motility and male fertility remains unclear.

Changes in phosphorylation levels of proteins such as AKAP4 (S245 and S447), C7orf61 (S34), KNG1 (S332), and RPL13 (S106) were also observed. AKAPs represent a family of anchoring proteins that function by binding the regulatory subunits of PKA and other enzymes to organelles or cytoskeleton elements, allowing the tight control of signal transduction in specific regions of the cell. Its phosphorylation, especially on Tyr residues, has been reported to be required during capacitation and acrosome reactions [21,49]. The phosphorylation of AKAP4 on Ser 447 was previously reported in capacitated human sperm [21]. We found that the phosphorylated levels of AKAP4 start to increase by age 30, possibly reflecting premature capacitation in this sperm population. The phosphorylation levels of RPL13 (S106) were also significantly upregulated in G1 compared with G4 (Figure 3). This ribosomal protein is phosphorylated at the S106 residue by AKT1 in leukemia cells [54] (Figure 4). However, how it is regulated and what function it has in sperm cells, especially in older men, is not known.

Gene Ontology and pathway enrichment analyses highlighted the significant involvement of the identified DEPPs in cellular stress responses. Among these pathways, both mitochondrial and endoplasmic reticulum unfolded protein response and heat shock response were previously recognized in human sperm following oxidative stress induction [33]. Furthermore, GO enrichment analysis revealed the involvement of DEPPs in the HSF1 activation pathway. HSF1 is a transcription factor that activates gene expression in response to several types of stress, including heat shock, oxidative stress, inflammation, and infection [55]. In response to cellular stress, HSF1 rapidly dissociates from chaperones such as HSPA1 and HSP90, is phosphorylated, and forms homotrimers that translocate into the nucleus, mediating the transcription of heat shock genes, such as HSP70, HSP90, and HSP27 [55]. Once transcribed, small heat shock proteins, like HSP27, form oligomeric structures that are phosphorylated and bind misfolded or unfolded proteins to correct their fold after stress [55]. Although the mechanisms of age-dependent reduction in male fertility remain largely unclear, oxidative stress has been hypothesized as the major cause of inducing DNA damage and disrupting protein homeostasis [18,56]. Thus, we evaluated the levels of some proteins involved in HSF1 activation, including HSF1, HSP90, HSP27, and phosphorylated HSP27 (Figure 5). Our results showed no correlation between men’s age and levels of HSF1 and HSP90. Previous studies showed increased levels of HSF1 mRNA in the sperm of men with varicocele and with oligozoospermia and that its protein levels significantly increased after oxidative stress induction [33]. On the other hand, HSP90 appears to be required for the maintenance of sperm motility and mitochondrial membrane potential in heat-stress conditions [57]. Despite a significant decrease in the levels of HSP27 with increasing age being observed, a tendency to increase its phosphorylation levels was shown, which suggests that the transcription of molecular chaperones such as HSP90 and HSP27 may not be occurring, possibly due to the absence of transcription and translation in sperm. Nevertheless, the cells cope with the stress by phosphorylating the available HSP27, resulting in functional oligomeric structures that correct protein folding in sperm induced by the aging process. This suggests a compensatory mechanism whereby sperm, which lack transcriptional activity, rely on post-translational modifications to manage proteotoxic stress.

While our findings provide critical insights into the molecular mechanisms of APA-related sperm dysfunction, several limitations must be acknowledged. The relatively small sample size, possible selection bias, and unmeasured environmental and lifestyle confounders limit the generalization of our results. Therefore, further studies should aim to validate these findings in larger, well-characterized cohorts across diverse populations. Additionally, assessing the functional implications of the identified proteins and phosphorylation changes will be essential prior to any clinical application of these molecules as biomarkers for the diagnosis and management of male infertility.

## 5. Conclusions

Although it is now accepted that APA negatively affects male fertility and reproductive outcomes, the knowledge on how it occurs is still scarce. Understanding the molecular changes in aging sperm can provide insight into the mechanisms underlying age-related male infertility and potentially inform new approaches to treatment and prevention.

This study provides compelling evidence that APA is associated with distinct molecular alterations in the human sperm proteome and phosphoproteome, even in the sperm fraction usually used in ART procedures. Through quantitative proteomics and phosphoproteomics, our findings revealed that age affects the protein expression and phosphorylation profile in human sperm, modulating signaling pathways crucial for normal sperm function, such as metabolism and stress response. In fact, the deregulated proteins mainly comprise molecular chaperones, cytoskeleton proteins, proteins linked to the fibrous sheath, and proteins associated with energy metabolism. Also, the identification of several proteins altered in the fraction of viable sperm of men older than 40 years, previously associated with altered semen parameters (e.g., AK1, PGK2) and fertilization success (e.g., LYZL1, PLCZ1), highlights the need to consider the age of the male partner in reproductive planning and clinical decision-making, especially when choosing the ART treatment used.

The molecular markers identified may be potentially used not only to explain situations of idiopathic infertility but also to clarify cases of failure in ART or repeated abortion associated with APA, overcoming the subjectivity of the conventional basic semen analysis. Ultimately, this work contributes to the growing body of evidence that male age should not be overlooked in fertility assessments, and it paves the way for future studies aimed at improving reproductive outcomes and offspring health through more personalized and age-informed approaches to male fertility care.

## Figures and Tables

**Figure 1 cells-14-00813-f001:**
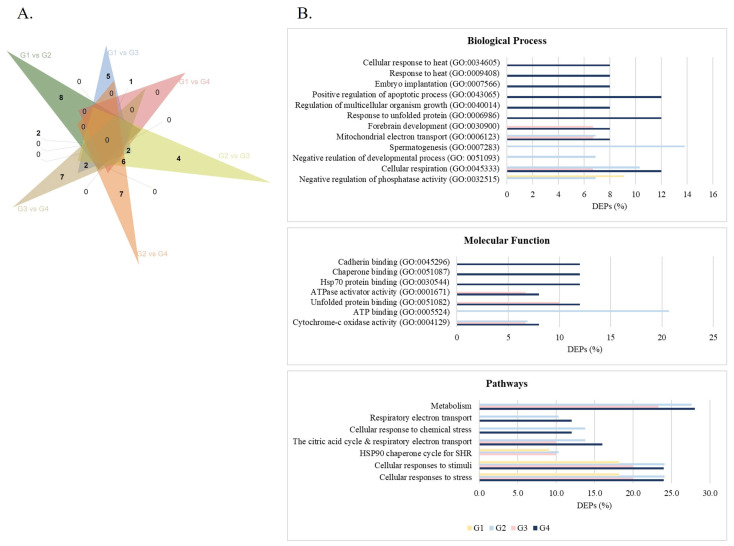
The differentially expressed proteins (DEPs) identified between the age groups. (**A**) Venn diagram displaying the number of DEPs in each group. (**B**) GO enrichment analyses for the DEPs identified in each age group showed significant (FDR < 0.05) biological processes, molecular functions, and pathways. Age groups: G1, ≤30 years old; G2, 31–35 years old; G3, 36–40 years old; G4, >40 years old.

**Figure 2 cells-14-00813-f002:**
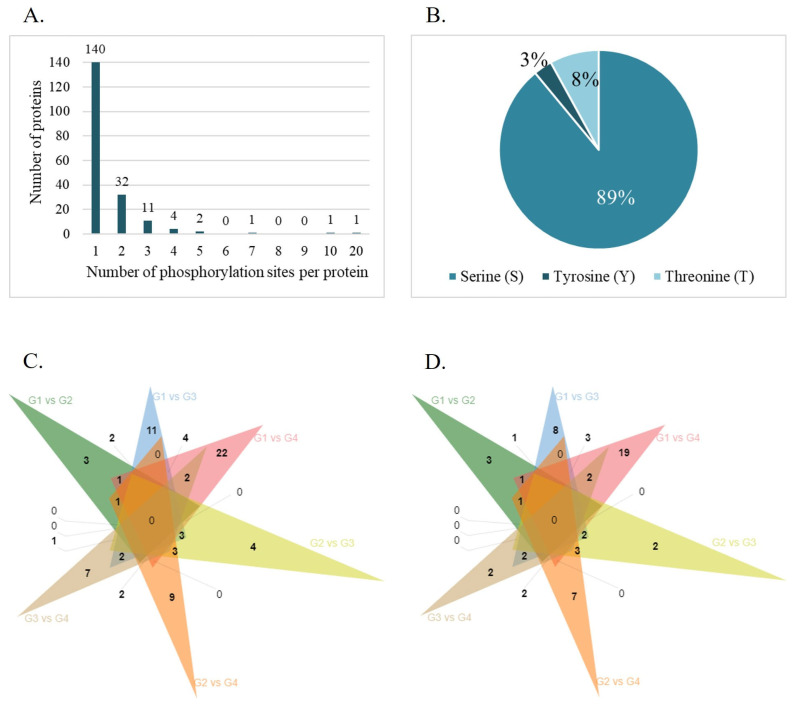
Identification of global phosphorylated proteins and their phosphorylated sites. (**A**) Number and distribution of phosphorylation sites of phosphorylated proteins identified. (**B**) The phosphorylation site distribution of serine (S), tyrosine (Y), and threonine (T). Venn diagram showing the number of common and exclusive differentially expressed phosphorylation sites (**C**) and proteins (**D**) between age groups (G1, ≤30 years old; G2, 31–35 years old; G3, 36–40 years old; G4, >40 years old).

**Figure 3 cells-14-00813-f003:**
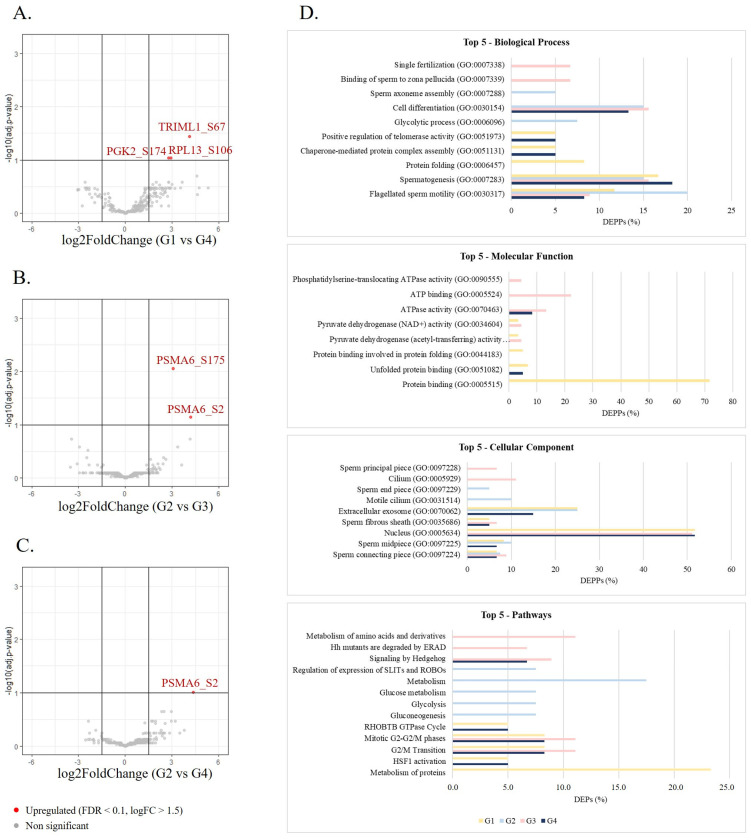
The identification of DEPPs between age groups. Volcano plots showing the most significant differentially expressed proteins (FDR < 0.1 and |log2FC| = 1.5) between G1 and G4 (**A**), G2 and G3 (**B**), and G3 and G4 (**C**). (**D**) GO enrichment analysis of DEPPs between groups. The top five significant (*p*-value < 0.05) biological processes, molecular functions, cellular components, and pathways in each group were represented. Age groups: G1, ≤30 years old; G2, 31–35 years old; G3, 36–40 years old; G4, >40 years old.

**Figure 4 cells-14-00813-f004:**
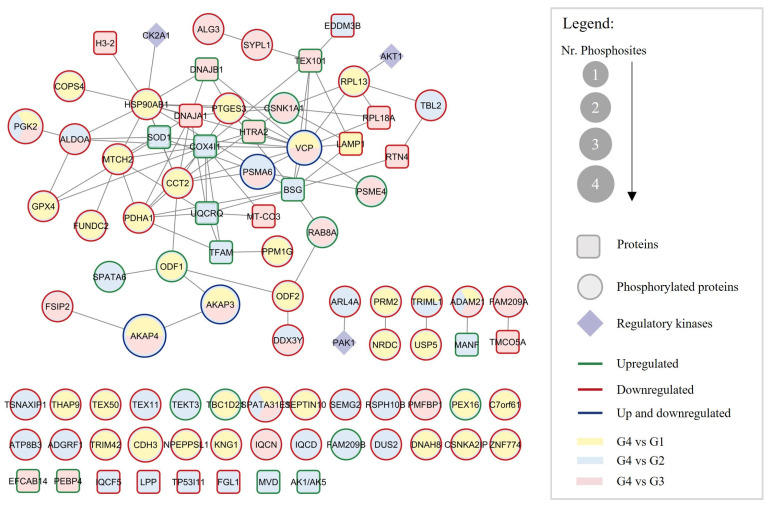
Protein–protein interaction network between the 25 DEPs (square nodes) and the 60 DEPPs (round nodes) identified in G4 and their regulatory kinases (purple diamond nodes). The green and red outlines represent DEPs upregulated (*n* = 13) and downregulated (*n* = 12), respectively. Concerning the DEPPS, the green, red, and dark blue outlines stand for DEPPs with phosphorylated sites upregulated (*n* = 9), downregulated (*n* = 47), and both up- and downregulated (*n* = 4) in G4, respectively. The yellow, blue, and pink round nodes represent DEPs and DEPPs in G4 compared with G1, G2, and G3, respectively. The round node size represents the number of phosphorylated sites altered. All proteins are represented by gene names. Age groups: G1, ≤30 years old; G2, 31–35 years old; G3, 36–40 years old; G4, >40 years old.

**Figure 5 cells-14-00813-f005:**
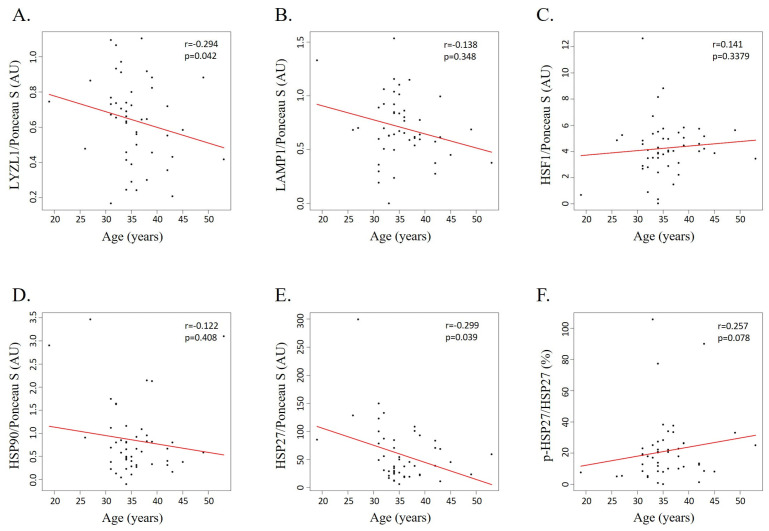
Scatterplots displaying the relationship between age and the levels of (**A**) LYZL1, (**B**) LAMP1, (**C**) HSF1, (**D**) HSP90, (**E**) HSP27, and (**F**) p-HSP27/HSP27. Ponceau S was used as the protein-loading control (*n* = 48).

**Table 1 cells-14-00813-t001:** The descriptive statistics of semen parameters by male age group and differences between groups. Data are expressed as mean ± standard deviation. Difference is significant at the 0.05 level. All samples used were normozoospermic and had complete liquefaction, normal appearance, and viscosity.

Parameter\Group	≤30 Years (G1; n = 66)	31–35 Years (G2; n = 96)	36–40 Years (G3; n = 100)	>40 Years (G4; n = 71)
Age (years)	25.0 ± 4.97	34.2 ± 0.84	37.8 ± 0.84	44.6 ± 5.94
Semen parameters				
Semen volume (mL)	4.375 ± 2.75	3.2 ± 1.2	3.8 ± 1.15	4.0 ± 1.84
Sperm concentration (10^6^/mL)	96.5 ± 99.07	81.0 ± 28.18	76.2 ± 67.59	61.0 ± 51.42
Total sperm counts (10^6^)	252.8 ± 147.24	269.8 ± 178.83	235.0 ± 103.79	189.4 ± 81.56
Total motility (%)	58.75 ± 7.80	75.0 ± 5.79	63.4 ± 5.94	65.6 ± 8.41
Progressive motility (%)	39.75 ± 6.5	58.8 ± 7.19	48.6 ± 4.93	47.4 ± 11.19
Non-progressive motility (%)	19.00 ± 5.83	16.2 ± 1.48	14.8 ± 4.76	14.2 ± 3.96
Immobility (%)	41.25 ± 7.80	25.00 ± 5.79	36.6 ± 5.94	38.4 ± 8.17
Morphological normal sperm (%)	7.0 ± 2.00	7.6 ± 1.52	6.75 ± 2.07	6.0 ± 1.73
Head defects (%)	84.0 ± 3.61	85.6 ± 4.34	86.8 ± 4.09	86.2 ± 7.26
Midpiece defects (%)	47.67 ± 2.31	43.8 ± 7.50	52.8 ± 9.44	51.2 ± 7.53
Principal piece defects (%)	23.67 ± 4.51	20.4 ± 6.43	22.2 ± 3.96	24.4 ± 5.13
Teratozoospermic index	1.67 ± 0.10	1.62 ± 0.11	1.75 ± 0.11	1.72 ± 0.14

**Table 2 cells-14-00813-t002:** The differentially expressed proteins (*p*-value < 0.05 and |log2FC| = 1.5) between the groups according to men’s age. Age groups: G1, ≤30 years old; G2, 31–35 years old; G3, 36–40 years old; G4, >40 years old. Legend: n.s., not significant.

Protein ID	Protein Name	Gene Name	*p*-Value	FDR	Log_2_ (Fold Change)
G1 vs. G2					
Q9NWV4	CXXC motif containing zinc binding protein	CZIB	0.019	n.s.	2.36
Q9BVA1	Tubulin beta-2B chain	TUBB2B	0.024	n.s.	2.31
P39210	Protein Mpv17	MPV17	0.023	n.s.	2.11
A4D1T9	Probable inactive serine protease 37	PRSS37	0.032	n.s.	1.98
Q15257	Serine/threonine-protein phosphatase 2A activator	PTPA	0.017	n.s.	1.74
P49841	Glycogen synthase kinase-3 beta	GSK3B	0.017	n.s.	1.71
Q9UJW0	Dynactin subunit 4	DCTN4	0.011	n.s.	1.69
Q9C099	Leucine-rich repeat and coiled-coil domain-containing protein 1	LRRCC1	0.016	n.s.	1.59
P55060	Exportin-2	CSE1L	0.045	n.s.	−1.83
P16444	Dipeptidase 1	DPEP1	0.039	n.s.	−2.1
G1 vs. G3					
P11279	Lysosome-associated membrane glycoprotein 1	LAMP1	0.047	n.s.	2.32
Q5VZ72	Izumo sperm-egg fusion protein 3	IZUMO3	0.029	n.s.	1.51
Q96S96	Phosphatidylethanolamine-binding protein 4	PEBP4	0.007	n.s.	2.18
Q96LI6	Heat shock transcription factor, Y-linked	HSFY2	0.016	n.s.	1.55
P31150	Rab GDP dissociation inhibitor alpha	GDI1	0.028	n.s.	1.55
Q9GZT6	Coiled-coil domain-containing protein 90B, mitochondrial	CCDC90B	0.025	n.s.	1.5
Q5H943	Kita-kyushu lung cancer antigen 1	CT83	0.039	n.s.	1.54
Q96M69	Leucine-rich repeat and guanylate kinase domain-containing protein	LRGUK	0.017	n.s.	1.84
Q6UWQ5	Lysozyme-like protein 1	LYZL1	0.020	n.s.	2.5
P31689	DnaJ homolog subfamily A member 1	DNAJA1	0.032	n.s.	−1.66
Q8N6Q1	Transmembrane and coiled-coil domain-containing protein 5A	TMCO5A	0.010	n.s.	−1.65
P00414	Cytochrome c oxidase subunit 3	MT-CO3	0.013	n.s.	−3.01
G1 vs. G4					
P11279	Lysosome-associated membrane glycoprotein 1	LAMP1	0.045	n.s.	2.35
G2 vs. G3					
Q9BVA1	Tubulin beta-2B chain	TUBB2B	0.010	n.s.	−2.58
P59666	Neutrophil defensin 3	DEFA3	0.041	n.s.	−3.45
P13073	Cytochrome c oxidase subunit 4 isoform 1, mitochondrial	COX4I1	0.048	n.s.	−1.87
Q96N23	Cilia- and flagella-associated protein 54	CFAP54	0.046	n.s.	−1.62
P00568	Adenylate kinase isoenzyme 1	AK1	0.002	n.s.	−2.2
P35613	Basigin	BSG	0.037	n.s.	−1.67
Q96M69	Leucine-rich repeat and guanylate kinase domain-containing protein	LRGUK	0.021	n.s.	1.67
Q6UWQ5	Lysozyme-like protein 1	LYZL1	0.012	n.s.	2.57
Q86YW0	1-phosphatidylinositol 4,5-bisphosphate phosphodiesterase zeta-1	PLCZ1	0.026	n.s.	1.58
P31689	DnaJ homolog subfamily A member 1	DNAJA1	0.035	n.s.	−1.54
O14683	Tumor protein p53-inducible protein 11	TP53I11	0.019	n.s.	1.69
Q93052	Lipoma-preferred partner	LPP	0.008	n.s.	1.73
P00414	Cytochrome c oxidase subunit 3	MT-CO3	0.041	n.s.	−2.25
O75935	Dynactin subunit 3	DCTN3	0.007	n.s.	1.84
A8MTL0	IQ domain-containing protein F5	IQCF5	0.028	n.s.	2.19
P16444	Dipeptidase 1	DPEP1	0.041	n.s.	1.96
G2 vs. G4					
P13073	Cytochrome c oxidase subunit 4 isoform 1, mitochondrial	COX4I1	0.018	n.s.	−2.29
Q00059	Transcription factor A, mitochondrial	TFAM	0.039	n.s.	−1.88
P00441	Superoxide dismutase [Cu-Zn]	SOD1	0.023	n.s.	−2.13
P00568	Adenylate kinase isoenzyme 1	AK1	0.002	n.s.	−2.18
P53602	Diphosphomevalonate decarboxylase	MVD	0.029	n.s.	−1.66
O14949	Cytochrome b-c1 complex subunit 8	UQCRQ	0.047	n.s.	−1.65
P55145	Mesencephalic astrocyte-derived neurotrophic factor	MANF	0.017	n.s.	−1.53
P35613	Basigin	BSG	0.005	n.s.	−2.35
Q08830	Fibrinogen-like protein 1	FGL1	0.004	n.s.	2.23
P56851	Epididymal secretory protein E3-beta	EDDM3B	0.036	n.s.	2.19
O14683	Tumor protein p53-inducible protein 11	TP53I11	0.018	n.s.	1.72
Q93052	Lipoma-preferred partner	LPP	0.003	n.s.	1.98
A8MTL0	IQ domain-containing protein F5	IQCF5	0.002	n.s.	3.38
G3 vs. G4					
Q96S96	Phosphatidylethanolamine-binding protein 4	PEBP4	0.007	n.s.	−2.09
P25685	DnaJ homolog subfamily B member 1	DNAJB1	0.019	n.s.	−1.52
Q02543	60S ribosomal protein L18a	RPL18A	0.004	n.s.	1.62
Q9BY14	Testis-expressed protein 101	TEX101	0.034	n.s.	−1.74
O75071	EF-hand calcium-binding domain-containing protein 14	EFCAB14	0.001	n.s.	−2.05
P31689	DnaJ homolog subfamily A member 1	DNAJA1	0.001	n.s.	2.6
O43464	Serine protease HTRA2, mitochondrial	HTRA2	0.015	n.s.	−1.99
Q5TEC6	Histone HIST2H3PS2	H3-2	0.024	n.s.	4.39
Q8N6Q1	Transmembrane and coiled-coil domain-containing protein 5A	TMCO5A	0.008	n.s.	1.61
Q9NQC3	Reticulon-4	RTN4	0.006	n.s.	1.7
P00414	Cytochrome c oxidase subunit 3	MT-CO3	0.034	n.s.	2.35

**Table 3 cells-14-00813-t003:** The differentially expressed phosphoproteins (DEPPs) between the groups according to men’s age. Age groups: G1, ≤30 years old; G2, 31–35 years old; G3, 36–40 years old; G4, >40 years old. Legend: n.s., not significant; P, phosphorylated; S, serine; Y, tyrosine; T, threonine.

Protein ID	Protein Name (Gene Name)	P-Sites	*p*-Value	FDR	Log2 (Fold Change)
G1 vs. G2					
P0C881	Radial spoke head 10 homolog B (RSPH10B)	S366	0.038	n.s.	−2.72
Q13200	26S proteasome non-ATPase regulatory subunit 2 (PSMD2)	S361	0.023	n.s.	−2.62
P60900	Proteasome subunit alpha type-6 (PSMA6)	S2	0.030	n.s.	−2.59
Q96RL7	Vacuolar protein sorting-associated protein 13A (VPS13A)	S1416	0.034	n.s.	−2.56
O14556	Glyceraldehyde-3-phosphate dehydrogenase, testis-specific (GAPDHS)	S282	0.002	n.s.	−2.36
Q5CZC0	Fibrous sheath-interacting protein 2 (FSIP2)	S6374	0.021	n.s.	−2.26
Q8IYX1	TBC1 domain family member 21 (TBC1D21)	S5	0.031	n.s.	−2.17
Q5JQC9	A-kinase anchor protein 4 (AKAP4)	S447	0.026	n.s.	−2.24
		S254	0.023	n.s.	2.61
Q5T9G4	Armadillo repeat-containing protein 12 (ARMC12)	T94	0.010	n.s.	1.6
P01042	Kininogen-1 (KNG1)	S332	0.035	n.s.	2.12
Q16563	Synaptophysin-like protein 1 (SYPL1)	S245	0.035	n.s.	2.29
P26373	60S ribosomal protein L13 (RPL13)	S106	0.002	n.s.	2.61
Q9NWH7	Spermatogenesis-associated protein 6 (SPATA6)	S265	0.002	n.s.	2.72
Q8IZ16	Uncharacterized protein C7orf61 (C7orf61)	S34	0.016	n.s.	3.3
G1 vs. G3					
Q5JX71	Protein FAM209A (FAM209A)	S141	0.005	n.s.	−3.29
Q92685	Dol-P-Man:Man(5)GlcNAc(2)-PP-Dol alpha-1,3-mannosyltransferase (ALG3)	S11	0.007	n.s.	−3.18
Q5JX69	Protein FAM209B (FAM209B)	S143	0.031	n.s.	−2.63
Q53SZ7	Proline-rich protein 30 (PRR30)	S188	0.030	n.s.	−2.49
Q8N4P6	Leucine-rich repeat-containing protein 71 (LRRC71)	S113	0.009	n.s.	−2.24
Q5JQC9	A-kinase anchor protein 4 (AKAP4)	S447	0.008	n.s.	−2.4
		Y303	0.006	n.s.	−2.38
		S254	0.028	n.s.	2.94
Q8TBY8	Polyamine-modulated factor 1-binding protein 1 (PMFBP1)	S765	0.044	n.s.	−2.39
Q5CZC0	Fibrous sheath-interacting protein 2 (FSIP2)	S6374	0.007	n.s.	−2.19
Q9UPU5	Ubiquitin carboxyl-terminal hydrolase 24 (USP24)	S2047	0.048	n.s.	−2.1
P60900	Proteasome subunit alpha type-6 (PSMA6)	S175	0.005	n.s.	1.52
Q9H0B3	IQ domain-containing protein N (IQCN)	S1057	0.048	n.s.	1.58
Q96JB1	Dynein heavy chain 8, axonemal (DNAH8)	S959	0.010	n.s.	1.91
Q5T9G4	Armadillo repeat-containing protein 12 (ARMC12)	S334	0.009	n.s.	1.91
		T94	0.001	n.s.	2.44
P29803	Pyruvate dehydrogenase E1 component subunit alpha, testis-specific form, mitochondrial (PDHA2)	S237	0.018	n.s.	2.19
Q96IV0	Peptide-N(4)-(N-acetyl-beta-glucosaminyl)asparagine amidase (NGLY1)	S94	0.043	n.s.	2.05
P26373	60S ribosomal protein L13 (RPL13)	S106	0.003	n.s.	2.41
P61006	Ras-related protein Rab-8A (RAB8A)	S185	0.036	n.s.	2.44
Q04837	Single-stranded DNA-binding protein, mitochondrial (SSBP1)	Y73	0.011	n.s.	2.46
P08559	Pyruvate dehydrogenase E1 component subunit alpha, somatic form, mitochondrial (PDHA1)	T231	0.044	n.s.	2.62
		S232	0.029	n.s.	2.72
		S291	0.004	n.s.	3.85
Q8N9V2	Probable E3 ubiquitin-protein ligase TRIML1 (TRIML1)	S67	0.004	n.s.	2.65
P49368	T-complex protein 1 subunit gamma (CCT3)	S252	0.039	n.s.	2.67
P08238	Heat shock protein HSP 90-beta (HSP90AB1)	S226	0.014	n.s.	2.7
Q9Y6C9	Mitochondrial carrier homolog 2 (MTCH2)	S114	0.014	n.s.	2.83
P01042	Kininogen-1 (KNG1)	S332	0.003	n.s.	3.12
Q9NTI2	Phospholipid-transporting ATPase IB (ATP8A2)	S46	0.040	n.s.	3.37
Q86UC2	Radial spoke head protein 3 homolog (RSPH3)	T243	0.016	n.s.	3.49
A6NJI9	Leucine-rich repeat-containing protein 72 (LRRC72)	S21	0.005	n.s.	3.98
Q5BJF6	Outer dense fiber protein 2 (ODF2)	S109	0.003	n.s.	4.19
Q8IZ16	Uncharacterized protein C7orf61 (C7orf61)	S34	0.002	n.s.	4.61
G1 vs. G4					
Q9Y5Y5	Peroxisomal membrane protein PEX16 (PEX16)	S183	0.018	n.s.	−2.31
Q14990	Outer dense fiber protein 1 (ODF1)	S5	0.008	n.s.	−2.18
O75969	A-kinase anchor protein 3 (AKAP3)	S170	0.026	n.s.	−2.08
Q8IYX1	TBC1 domain family member 21 (TBC1D21)	S5	0.049	n.s.	−1.96
Q5JQC9	A-kinase anchor protein 4 (AKAP4)	Y303	0.026	n.s.	−1.92
		S447	0.032	n.s.	−1.91
		S254	0.006	n.s.	3.35
P36969	Phospholipid hydroperoxide glutathione peroxidase (GPX4)	S40	0.029	n.s.	1.52
O15355	Protein phosphatase 1G (PPM1G)	S517	0.035	n.s.	1.57
Q9H5L6	DNA transposase THAP9 (THAP9)	S87	0.048	n.s.	1.61
Q8IWZ5	Tripartite motif-containing protein 42 (TRIM42)	S123	0.037	n.s.	1.74
A6NEC2	Puromycin-sensitive aminopeptidase-like protein (NPEPPSL1)	S39	0.010	n.s.	1.76
Q6NX45	Zinc finger protein 774 (ZNF774)	S192	0.047	n.s.	1.78
Q96JB1	Dynein heavy chain 8, axonemal (DNAH8)	S959	0.007	n.s.	1.83
Q9P0V9	Septin-10 (SEPTIN10)	S214	0.026	n.s.	1.85
Q9UKJ8	Disintegrin and metalloproteinase domain-containing protein 21 (ADAM21)	S670	0.019	n.s.	1.94
Q9Y6C9	Mitochondrial carrier homolog 2 (MTCH2)	S114	0.048	n.s.	2.08
Q9BT78	COP9 signalosome complex subunit 4 (COPS4)	S297	0.018	n.s.	2.09
P45974	Ubiquitin carboxyl-terminal hydrolase 5 (USP5)	S783	0.046	n.s.	2.13
Q15185	Prostaglandin E synthase 3 (PTGES3)	S82	0.044	n.s.	2.23
Q9BWH2	FUN14 domain-containing protein 2 (FUNDC2)	S151	0.042	n.s.	2.25
A0A1B0GTY4	Testis-expressed protein 50 (TEX50)	S142	0.022	n.s.	2.27
Q6ZUB1	Spermatogenesis-associated protein 31E1 (SPATA31E1)	S462	0.045	n.s.	2.31
O43847	Nardilysin (NRDC)	S94	0.042	n.s.	2.33
A0A1B0GTH6	Casein kinase II subunit alpha-interacting protein (CSNKA2IP)	S212	0.033	n.s.	2.35
P07205	Phosphoglycerate kinase 2 (PGK2)	S175	0.012	n.s.	2.42
		S174	0.001	0.092	3.13
P08238	Heat shock protein HSP 90-beta (HSP90AB1)	S226	0.011	n.s.	2.75
P26373	60S ribosomal protein L13 (RPL13)	S106	0.001	0.092	2.77
P01042	Kininogen-1 (KNG1)	S332	0.008	n.s.	2.77
P55072	Transitional endoplasmic reticulum ATPase (VCP)	S13	0.021	n.s.	2.8
P78371	T-complex protein 1 subunit beta (CCT2)	S470	0.033	n.s.	2.89
P08559	Pyruvate dehydrogenase E1 component subunit alpha, somatic form, mitochondrial (PDHA1)	S291	0.034	n.s.	2.91
P04554	Protamine-2 (PRM2)	S10	0.034	n.s.	3.53
Q5BJF6	Outer dense fiber protein 2 (ODF2)	S109	0.007	n.s.	3.71
Q8IZ16	Uncharacterized protein C7orf61 (C7orf61)	S34	0.003	n.s.	4.01
Q8N9V2	Probable E3 ubiquitin-protein ligase TRIML1 (TRIML1)	S67	0.000	0.036	4.17
P22223	Cadherin-3 (CDH3)	T741	0.020	n.s.	5.32
		T737	0.028	n.s.	4.74
G2 vs. G3					
Q5JX71	Protein FAM209A (FAM209A)	S141	0.024	n.s.	−3.91
Q5JX69	Protein FAM209B (FAM209B)	S143	0.003	n.s.	−3.64
Q5JQC9	A-kinase anchor protein 4 (AKAP4)	S536	0.044	n.s.	−3.1
Q16563	Synaptophysin-like protein 1 (SYPL1)	S245	0.005	n.s.	−3
Q8N4P6	Leucine-rich repeat-containing protein 71 (LRRC71)	S113	0.048	n.s.	−1.53
Q9H0B3	IQ domain-containing protein N (IQCN)	S1171	0.012	n.s.	−2.47
		S1057	0.033	n.s.	1.78
		T720	0.012	n.s.	2.43
Q92685	Dol-P-Man:Man(5)GlcNAc(2)-PP-Dol alpha-1,3-mannosyltransferase (ALG3)	S11	0.030	n.s.	−2.34
Q9NWH7	Spermatogenesis-associated protein 6 (SPATA6)	S265	0.007	n.s.	−2.21
O60423	Phospholipid-transporting ATPase IK (ATP8B3)	S891	0.022	n.s.	1.52
Q96JB1	Dynein heavy chain 8, axonemal (DNAH8)	S959	0.015	n.s.	1.53
Q17R55	Protein FAM187B (FAM187B)	S362	0.049	n.s.	1.85
Q96M32	Adenylate kinase 7 (AK7)	S61	0.026	n.s.	1.99
Q9Y6C9	Mitochondrial carrier homolog 2 (MTCH2)	S114	0.016	n.s.	2.57
P60900	Proteasome subunit alpha type-6 (PSMA6)	S175	0.000	0,0088	2.74
		S2	0.000	0,0726	3.89
Q2TAA8	Translin-associated factor X-interacting protein 1 (TSNAXIP1)	T318	0.005	n.s.	3.26
Q5T601	Adhesion G-protein coupled receptor F1 (ADGRF1)	S871	0.033	n.s.	3.6
A6NJI9	Leucine-rich repeat-containing protein 72 (LRRC72)	S21	0.002	n.s.	4.16
G2 vs. G4					
Q5JX69	Protein FAM209B (FAM209B)	S143	0.033	n.s.	−2.43
Q9NWH7	Spermatogenesis-associated protein 6 (SPATA6)	S265	0.008	n.s.	−2.22
Q9BXF9	Tektin-3 (TEKT3)	S64	0.027	n.s.	−2.14
P04075	Fructose-bisphosphate aldolase A (ALDOA)	S39	0.048	n.s.	1.61
Q9Y4P3	Transducin beta-like protein 2 (TBL2)	S298	0.009	n.s.	1.65
O60423	Phospholipid-transporting ATPase IK (ATP8B3)	S891	0.016	n.s.	1.7
Q96DY2	Dynein regulatory complex protein 10 (IQCD)	S133	0.009	n.s.	1.94
P07205	Phosphoglycerate kinase 2 (PGK2)	S174	0.003	n.s.	2.35
Q2TAA8	Translin-associated factor X-interacting protein 1 (TSNAXIP1)	T318	0.011	n.s.	2.81
Q6ZUB1	Spermatogenesis-associated protein 31E1 (SPATA31E1)	S1057	0.012	n.s.	2.83
Q02383	Semenogelin-2 (SEMG2)	S352	0.046	n.s.	3.18
Q8N9V2	Probable E3 ubiquitin-protein ligase TRIML1 (TRIML1)	S67	0.002	n.s.	3.41
P60900	Proteasome subunit alpha type-6 (PSMA6)	S2	0.000	0,099	4.21
Q9UKJ8	Disintegrin and metalloproteinase domain-containing protein 21 (ADAM21)	S670	0.003	n.s.	2.48
P40617	ADP-ribosylation factor-like protein 4A (ARL4A)	S143	0.023	n.s.	2.3
Q8IYF3	Testis-expressed protein 11 (TEX11)	S367	0.044	n.s.	1.82
Q9NX74	tRNA-dihydrouridine(20) synthase [NAD(P)+]-like (DUS2)	T369	0.016	n.s.	1.94
O15523	ATP-dependent RNA helicase DDX3Y (DDX3Y)	S408	0.041	n.s.	2.32
Q5T601	Adhesion G-protein coupled receptor F1 (ADGRF1)	S867	0.044	n.s.	3.49
		S871	0.026	n.s.	3.78
P0C881	Radial spoke head 10 homolog B (RSPH10B)	S366	0.041	n.s.	2.51
G3 vs. G4					
O75969	A-kinase anchor protein 3 (AKAP3)	S170	0.006	n.s.	−3.37
		S176	0.013	n.s.	−2.75
		S208	0.046	n.s.	2.47
P48729	Casein kinase I isoform alpha (CSNK1A1)	S4	0.008	n.s.	−3.33
Q14997	Proteasome activator complex subunit 4 (PSME4)	S293	0.003	n.s.	−3.18
Q5JQC9	A-kinase anchor protein 4 (AKAP4)	S817	0.032	n.s.	−2.17
P61006	Ras-related protein Rab-8A (RAB8A)	S185	0.047	n.s.	−2.16
P55072	Transitional endoplasmic reticulum ATPase (VCP)	S3	0.024	n.s.	−1.98
P60900	Proteasome subunit alpha type-6 (PSMA6)	S175	0.005	n.s.	−1.63
Q6ZUB1	Spermatogenesis-associated protein 31E1 (SPATA31E1)	S462	0.032	n.s.	1.52
P04075	Fructose-bisphosphate aldolase A (ALDOA)	S39	0.028	n.s.	1.52
P07205	Phosphoglycerate kinase 2 (PGK2)	S174	0.014	n.s.	1.83
Q16563	Synaptophysin-like protein 1 (SYPL1)	S245	0.015	n.s.	2.28
Q8TBY8	Polyamine-modulated factor 1-binding protein 1 (PMFBP1)	S765	0.015	n.s.	2.5
Q5CZC0	Fibrous sheath-interacting protein 2 (FSIP2)	S3641	0.001	n.s.	2.55
Q9H0B3	IQ domain-containing protein N (IQCN)	S1171	0.001	n.s.	3.23
Q5JX71	Protein FAM209A (FAM209A)	S141	0.002	n.s.	6.95
Q9UKJ8	Disintegrin and metalloproteinase domain-containing protein 21 (ADAM21)	S670	0.046	n.s.	1.53
Q92685	Dol-P-Man:Man(5)GlcNAc(2)-PP-Dol alpha-1,3-mannosyltransferase (ALG3)	S11	0.004	n.s.	3.23
O15523	ATP-dependent RNA helicase DDX3Y (DDX3Y)	S408	0.047	n.s.	2.25

## Data Availability

The data underlying this article are available in the article and in its online Appendix A.

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
