# Peer review of "Advanced Paternal Age and Sperm Proteome Dynamics: A Possible Explanation for Age-Associated Male Fertility Decline"

_cells, 2025, doi:10.3390/cells14110813_

Round 1
Reviewer 1 Report
Comments and Suggestions for Authors
Main Issues and Limitations
- Small sample size and group stratification
Only 19 samples were profiled by MS. The sample size is still too small.
The validation cohort partially addresses this but remains modest and unbalanced across age and semen‐parameter strata.
- Volunteers were drawn from fertility clinic attendees, not from the general population; they may overrepresent subfertile men or specific lifestyles.
Environmental and lifestyle factors were not controlled or adjusted in multivariate analyses, yet these can strongly influence sperm proteome and phosphorylation.
- Limited functional validation
Only two proteins and were validated by slot blot—and correlations with age were weak.
No assays of sperm function were performed in the same samples to link molecular changes to physiological outcomes.
4. The discussion on the results still needs to be more rigorous.
If the above problems cannot be properly addressed, I believe this paper cannot be accepted.
Author Response
Comment 1: Main Issues and Limitations
Response 1: Thank you for your thoughtful comments and suggestions. We have revised the manuscript accordingly, and the changes are reflected in the track changes version. Below, we provide a detailed point-by-point response to each of your comments.
Comment 2: Small sample size and group stratification. Only 19 samples were profiled by MS. The sample size is still too small. The validation cohort partially addresses this, but remains modest and unbalanced across age and semen‐parameter strata.
Response 2: We may concur that the sample is small; however, recruiting healthy donors with strictly normal semen parameters, particularly in the younger (<25 years) and older (>40 years) age groups, is notably challenging and constrained our study design. We have clearly stated this as a limitation in the revised manuscript (lines 648-662) – “Although our study revealed a large number of proteins and phosphorylated residues significantly affected by age in the fraction of sperm usually used in ART treatments, the small sample size, possible selection bias and the influence of confounders, like environmental and lifestyle factors, constitute the main limitations of this work”. Regarding the validation cohort, while we included a broader sample regardless of semen parameters, all samples underwent density gradient centrifugation, and only the highly motile, viable sperm fraction—typically used in ART—was analyzed. Since this selection process inherently excludes immotile and dead sperm, we did not stratify based on semen quality, aligning with clinical relevance for ART procedures.
Comment 2: Volunteers were drawn from fertility clinic attendees, not from the general population; they may overrepresent subfertile men or specific lifestyles. Environmental and lifestyle factors were not controlled or adjusted in multivariate analyses, yet these can strongly influence sperm proteome and phosphorylation.
Response 2: While our volunteers were indeed recruited from fertility consultations at the local hospital, we implemented strict exclusion criteria to reduce potential confounding. As detailed in the Methods section (lines 103–107), we excluded individuals with known reproductive conditions, those undergoing medication, and lifestyle factors such as smoking or heavy alcohol consumption. These measures aimed to minimize the influence of external factors on the sperm proteome. Nonetheless, we recognize the possibility of residual confounding from undetected idiopathic infertility or uncontrolled lifestyle/environmental factors. We acknowledge this as a limitation. However, because we focused on the sperm fraction routinely used in ART, these factors may have a reduced impact on the subset analyzed, supporting the clinical relevance of our findings.
Comment 3: Limited functional validation. Only two proteins and were validated by slot blot—and correlations with age were weak. No assays of sperm function were performed in the same samples to link molecular changes to physiological outcomes.
Response 3: The two proteins selected for validation were chosen based on antibody availability and their potential relevance to male fertility and fertilization. We recognize the limitations of slot blotting as a semi-quantitative technique, particularly when compared to the sensitivity and specificity of mass spectrometry (MS). MS allows precise quantification of peptides, detection of isoforms, and assessment of post-translational modifications, while immunoblotting techniques like slot blot are limited by antibody specificity and epitope accessibility, which can be affected by protein conformation or modification. These technical differences likely contribute to the weaker observed correlations. Although some researchers argue that MS-based proteomics may not require external validation, we opted to include slot blot analysis for illustrative purposes. Despite its limitations, the observed trends were consistent with our MS data, providing an additional, albeit modest, layer of validation. We acknowledge that additional functional assays would further strengthen our conclusions, and it represent an avenue for future work.
Comment 4: The discussion on the results still needs to be more rigorous.
Response 4: We have revised and expanded the discussion section, to better discuss the results and contextualize our findings with the existing literature. These changes are detailed in the track changes version of the manuscript.
Reviewer 2 Report
Comments and Suggestions for Authors
Authors of the original article „Advanced paternal age and sperm proteome dynamics: a possible explanation for age-associated male fertility decline” represent changes in some proteins expression correlating with males age. Although in the presented work some of proteins were identified that show differential expression in different age groups of men, in my opinion there are several points of the work that require improvement.
The first thing that draws attention in this publication is the very small group of patients taken into the study. Additionally, the authors did not select equal numbers of patients for the groups. It is difficult to make reliable statistics in such a case (especially to draw the average and deviation). I think that a better solution would be to increase the number of patients in each age group to 10 and provide the median value and the range of 25-75% in the obtained results. Therefore, when making statistics on such a small group, it is difficult to draw binding conclusions regarding the differences between the expression of individual proteins, especially since such a correlation was not confirmed in the later slot blot analysis. I do not know if the authors have this type of material, but it would be good to have a group of men older than 50 years and increased the number of patients in the remaining groups, perhaps then we would obtain greater differences.
Why did the authors choose such proteins to confirm in slot blot? I would say LAMP1 and LYZL1 are accurate. Why authors didn’t choose some of proteins identified in MS with the lowest p-value, for example AK1, which expression is different in two correlations (G2 v G3 and G3 v G4), and which p-value is very low.
In the Slot-Blot chapter, the authors described the standardization using the Ponceau S. Ponceau S is used to check the success of the electrotransfer or dot to membrana, but it cannot be used for standardization because the staining is very impermanent. I suggest the authors to think about another strategy, e.g. pooling a group of patients from a given age compartment and conducting standardization in relation to its activity or staining the proteinogram on the slot slot with a permanent dye more durable than Ponceau S, e.g. silver.
Lines 453 - 454. "Forty-six DEPs between the groups were identified (Table 2), 25 of them with altered expression in men with APA (G4) compared with the other groups (Figure 1)." This is not clear. Table 2 - I count 47 DEPs, but please check. Where are 25 with altered expression on Fig1? Please make it more clear.
In the line 480, it should be LYZL1 (there is LIZL1).
Line 530 - there is Figure 3G, and on the figure 3 there is no "G", please check and unify the markings.
Author Response
Comment 1: Authors of the original article „Advanced paternal age and sperm proteome dynamics: a possible explanation for age-associated male fertility decline” represent changes in some proteins expression correlating with males age. Although in the presented work some of proteins were identified that show differential expression in different age groups of men, in my opinion there are several points of the work that require improvement.
Response 1: Thank you for your insightful comments and constructive suggestions. We have revised the manuscript to address your points, and the changes are reflected in the tracked version. Below, we provide a detailed point-by-point response to each of your comments.
Comment 2: The first thing that draws attention in this publication is the very small group of patients taken into the study. Additionally, the authors did not select equal numbers of patients for the groups. It is difficult to make reliable statistics in such a case (especially to draw the average and deviation). I think that a better solution would be to increase the number of patients in each age group to 10 and provide the median value and the range of 25-75% in the obtained results. I do not know if the authors have this type of material, but it would be good to have a group of men older than 50 years and increased the number of patients in the remaining groups, perhaps then we would obtain greater differences.
Response 2: We acknowledge the limitation of a small sample size in this study. Recruiting healthy donors with strictly normal semen parameters, particularly in the younger (<25 years) and older (>40 years) age groups, has proven difficult, limiting our ability to expand the sample. We have stated this as a limitation in the revised manuscript (lines 648-662) – “Although our study revealed a large number of proteins and phosphorylated residues significantly affected by age in the fraction of sperm usually used in ART treatments, the small sample size, possible selection bias and the influence of confounders, like environmental and lifestyle factors, constitute the main limitations of this work”. ). Unfortunately, we do not currently have access to additional samples needed to increase the number of patients per group, nor to include a group of men over 50, as such patients are rarely seen in fertility consultations at our hospital.
Comment 3: When making statistics on such a small group, it is difficult to draw binding conclusions regarding the differences between the expression of individual proteins, especially since such a correlation was not confirmed in the later slot blot analysis.
Response 3: Regarding the discrepancies observed between the MS and slot blot results, we believe these may be partly attributed to the differences between the two techniques. Slot blotting is a semi-quantitative method, which is inherently less sensitive and specific than mass spectrometry (MS). MS offers precise quantification of peptides, detection of isoforms, and the ability to analyze post-translational modifications, while immunoblotting techniques like slot blot are limited by factors such as antibody specificity and epitope accessibility. These limitations may explain the weaker correlations observed. While some researchers argue that MS-based proteomics does not require further validation, we included slot blot analysis as an additional layer of validation, albeit with the understanding that it has limitations. Despite these limitations, the trends observed by slot blot analysis were consistent with the MS data, providing supplementary evidence.
Comment 4: Why did the authors choose such proteins to confirm in slot blot? I would say LAMP1 and LYZL1 are accurate. Why authors didn’t choose some of proteins identified in MS with the lowest p-value, for example AK1, which expression is different in two correlations (G2 v G3 and G3 v G4), and which p-value is very low.
Response 4: The proteins chosen for validation (LAMP1 and LYZL1) were selected based on their relevance to male fertility and the availability of suitable antibodies. As stated in lines 480-490, both LAMP1 and LYZL1 are located in the acrosome of mammalian sperm and may play crucial roles in the acrosome reaction and the digestion of the zona pellucida—key processes in fertilization. While AK1 could have been validated, we prioritized proteins that have greater biological relevance to the mechanisms of fertilization and infertility, particularly in the context of our study population. We believe this focus will provide deeper insights into the underlying causes of infertility in these men.
Comment 5: In the Slot-Blot chapter, the authors described the standardization using the Ponceau S. Ponceau S is used to check the success of the electrotransfer or dot to membrana, but it cannot be used for standardization because the staining is very impermanent. I suggest the authors to think about another strategy, e.g. pooling a group of patients from a given age compartment and conducting standardization in relation to its activity or staining the proteinogram on the slot slot with a permanent dye more durable than Ponceau S, e.g. silver.
Response 5: We used Ponceau S for normalization purposes to correct protein loading differences, which can otherwise obscure real expression differences. However, we also performed protein quantification using the BCA assay and loaded an equal amount of protein in each slot to mitigate this issue. Ponceau S is widely used for total protein staining and normalization in immunoblotting (doi: 10.1016/j.ab.2019.03.010), as it helps visualize all proteins transferred to the membrane, allowing for adjustments for variations in loading or experimental error. While we acknowledge that Ponceau S is impermanent and easily removed, it was specifically chosen for its utility in this context, particularly when the blot needs to be reused for another detection method, such as immunoblotting.
Comment 6: Lines 453 - 454. "Forty-six DEPs between the groups were identified (Table 2), 25 of them with altered expression in men with APA (G4) compared with the other groups (Figure 1)." This is not clear. Table 2 - I count 47 DEPs, but please check. Where are 25 with altered expression on Fig1? Please make it more clear.
Response 6: We have carefully re-examined the number of DEPs and confirmed that 46 DEPs were identified, after excluding duplicate proteins (proteins identified as DEPs in more than one comparison were counted only once). Additionally, we have revised the text to reflect that the 25 DEPs with altered expression in the G4 group are most clearly shown in Figure 4, not Figure 1. We have made the necessary corrections for clarity.
Comment 7: In the line 480, it should be LYZL1 (there is LIZL1).
Response 7: The correction has been made.
Comment 8: Line 530 - there is Figure 3G, and on the figure 3 there is no "G", please check and unify the markings.
Response 8: We have corrected this error. Figure 3G should be Figure 3C. The labels and figure references have been reviewed and unified throughout the manuscript.
Round 2
Reviewer 1 Report
Comments and Suggestions for Authors
no more problem
Reviewer 2 Report
Comments and Suggestions for Authors
Thanks to the authors for their comprehensive explanations of the comments on methodology. I think that the article can now be accepted for publication and will contribute to the expansion of knowledge on male age-related infertility.